# Nomogram-Based Survival Predictions and Treatment Recommendations for Locally Advanced Esophageal Squamous Cell Carcinoma Treated with Upfront Surgery

**DOI:** 10.3390/cancers14225567

**Published:** 2022-11-13

**Authors:** Jie Zhu, Yongtao Han, Wenjie Ni, Xiao Chang, Lei Wu, Yi Wang, Li Jiang, Yan Tan, Zefen Xiao, Qifeng Wang, Lin Peng

**Affiliations:** 1Sichuan Cancer Center, Radiation Oncology Key Laboratory of Sichuan Province, Department of Radiation Oncology, Sichuan Cancer Hospital & Institute, School of Medicine, University of Electronic Science and Technology of China, Chengdu 610042, China; 2Sichuan Cancer Center, Department of Thoracic Surgery, Sichuan Cancer Hospital & Institute, School of Medicine, University of Electronic Science and Technology of China, Chengdu 610042, China; 3Department of Radiation Oncology, Beijing Shijitan Hospital, Capital Medical University, Beijing 100038, China; 4National Cancer Center/National Clinical Research Center for Cancer/Cancer Hospital, Departments of Radiation Oncology, Chinese Academy of Medical Sciences and Peking Union Medical College, Beijing 100021, China

**Keywords:** locally advanced, esophageal squamous cell carcinoma, esophagectomy, nomogram, adjuvant treatment

## Abstract

**Simple Summary:**

In China, upfront surgery is still currently common in clinical practice for locally advanced esophageal squamous cell carcinoma (LA-ESCC). The aim of this study is to develop a prognostic nomogram, quantify survival benefit, and guide risk-dependent adjuvant therapy in LA-ESCCs treated with upfront surgery. A nomogram was successfully established with high accuracy through modeling with single-center, large-scale retrospective data. Comprehensive validation was performed internally and externally. Survival improvement from adjuvant therapy was quantified and plotted corresponding to nomogram score, and at least 10% improvement in 5-year OS attributing to adjuvant chemoradiotherapy and chemotherapy was expected in almost all patients and patients mainly with high-intermediate risk, respectively.

**Abstract:**

Background and purpose: The aim of this study is to develop a prognostic nomogram, quantify survival benefit, and guide risk-dependent adjuvant therapy for locally advanced esophageal squamous cell carcinoma (LA-ESCC) after esophagectomy. Materials and methods: This was a single-center, retrospective study of consecutive LA-ESCCs treated by curative-intent esophagectomy with internal validation and independent external validation in a randomized controlled trial. After factor selection by the least absolute shrinkage and selection operator regression, a nomogram was developed to estimate 5-year overall survival (OS) based on the Cox proportional hazards model. The area under the curve (AUC) and calibration plot were used to determine its discriminative and predictive capacities, respectively. Survival improvement from adjuvant therapy was quantified and plotted corresponding to nomogram score. Results: A total of 1077, 718, and 118 patients were included for model development, internal validation, and external validation, respectively. The nomogram identified eight significant prognostic factors: gender, pathological T and N stages, differentiation, surgical margin, lymphovascular invasion, number of lymph node resection, and adjuvant therapy. The nomogram showed superior discriminative capacity than TNM stage (AUC: 0.76 vs. 0.72, *p* < 0.01), with significant survival differences among different risk stratifications. The calibration plot illustrated a good agreement between nomogram-predicated and actual 5-year OS. Consistent results were concluded after external validation. At least 10% 5-year OS improvement from adjuvant chemoradiotherapy and chemotherapy was expected in almost all patients (nomogram score 110 to 260) and patients mainly with high-intermediate risk (nomogram score 159 to 207), respectively. Conclusions: The clinicopathological nomogram predicting 5-year OS for LA-ESCC after esophagectomy was developed with high accuracy. The proposed nomogram showed better performance than TNM stage and provided risk-dependent and individualized adjuvant treatment recommendations.

## 1. Introduction

Esophageal carcinoma (EC) is a major malignancy worldwide, ranking seventh in incidence and sixth in mortality, respectively [1]. In China, EC is the third most common cancer, with an estimate of 477,900 new cases and 375,000 deaths annually [2]. Although neoadjuvant chemoradiotherapy followed by radical esophagectomy significantly improves survival and is recommended as the standard therapy by the National Comprehensive Cancer Network guideline, the long-term survival of locally advanced EC is still far from satisfaction [3]. Actually, a large proportion of Chinese patients with locally advanced EC receive surgery first, and radical esophagectomy plus adjuvant chemoradiotherapy is an alternative real-world multimodal treatment pattern in China [4,5,6,7,8]. In the current neoadjuvant era, the exploration on postoperative survival predictions and adjuvant therapy options is still too early to be judged as obsolete.

There is significant survival heterogeneity among patients with the same disease stage after esophagectomy; therefore, it is difficult to acquire precise survival predictions and recommend risk-adaptive postoperative treatments depending on TNM stage alone [9]. Besides disease stage, many other clinicopathological factors are independently associated with survival, such as gender, tumor location, lymphovascular invasion, differentiation grade, and adjuvant therapy [9,10]. It is urgent to identify a novel clinicopathological risk-stratification model beyond the current TNM stage, which includes comprehensive survival-associated factors, to stratify patients into different risk levels, offer individualized survival predictions, and guide risk-dependent adjuvant treatments. 

The visual format of nomogram can provide a statistical prediction model that is easily understood by both oncologists and patients. In accordance with this, the development of nomograms to quantify risk in some malignancies has improved predictive accuracy for clinical outcomes compared with conventional risk stratifications [11,12,13,14]. Some nomograms predicting survival of locally advanced EC after esophagectomy based on large-scale populations have been developed and have provided useful information on survival prediction [9,10,15,16,17]. However, risk-dependent adjuvant therapies were rarely recommended to individuals in previously reported nomograms. The aim of this study is to develop and validate a clinicopathological nomogram for overall survival (OS) in locally advanced esophageal squamous cell carcinoma (LA-ESCC) after esophagectomy and then recommend risk-dependent and individualized adjuvant therapies based on nomogram score.

## 2. Materials and Methods

### 2.1. Study Population

The eligibility criteria were: (1) pathologically confirmed thoracic LA-ESCC; (2) standard McKeown or Ivor Lewis esophagectomy with curative intention; (3) pathological stage T3-4a for tumor or N1-3 for regional lymph node with no distant metastasis (pT3-4aN0M0, pT1-4aN1-3M0) in 8th AJCC TNM stage; (4) adjuvant therapy or observation; and (5) adequate renal, hepatic, and bone marrow functions. Patients were excluded if they met any of the followings: (1) cervical EC; (2) pathologically confirmed adenocarcinoma, neuroendocrine carcinoma, or small cell carcinoma; (3) palliative esophagectomy; (4) any neoadjuvant therapy; (5) adjuvant radiotherapy alone; (6) unqualified surgical resection by experts’ review; and (7) missing data on the variables of interest. Particularly, we excluded adjuvant radiotherapy in this study because the number of patients receiving adjuvant radiotherapy (*n* = 40) was not enough for precise modeling. Salvage treatment after disease progression was allowed. 

We retrospectively identified consecutive patients who underwent curative-intent esophagectomy in Sichuan Cancer Hospital & Institute, Chengdu, China, from January 2008 to December 2017. The data of Sichuan Cancer Hospital were randomly divided into primary cohort and internal validation cohort with a ratio of 3:2. External validation was performed using an independent prospective phase III randomized controlled trial (RCT) (NCT02279134) held in National Cancer Center/Cancer Hospital, Chinese Academy of Medical Sciences, Beijing, China, in which patients with LA-ESCC (pathological stage IIB to III, 7th AJCC TNM stage [18]) were randomly assigned to esophagectomy alone, esophagectomy plus adjuvant radiotherapy, or esophagectomy plus adjuvant chemoradiotherapy using a computer-generated random number code method [4]. The data of patients receiving esophagectomy alone or esophagectomy plus adjuvant chemoradiotherapy in the RCT were used for independent external validation. All patients were restaged using 8th AJCC TNM stage [19].

Written informed consents were obtained from all patients prior to treatment. This study was approved by the Institutional Review Board of Sichuan Cancer Hospital and Institute (SCCHEC-02-2020-015) and the Ethics Committee of Cancer Institute and Hospital, Chinese Academy of Medical Sciences, Beijing (14-090/880). This study was performed in accordance with the principles of Declaration of Helsinki. This study followed the checklist on transparent reporting of a multivariable prediction model for individual prognosis or diagnosis (TRIPOD) (Appendix A).

### 2.2. Data Extraction, Evaluation, and Follow-Up

This data set included patient demographics (age and sex), Karnofsky performance status (KPS), pathologic characteristics (location, length, stage for tumor and lymph node, differentiation, surgical margin, lymphovascular invasion, nerve invasion, and number of lymph node resection), adjuvant chemotherapy or chemoradiotherapy, and follow-up data (follow-up duration and survival). Continuous variables were categorized according to clinical reasoning or statistical methods and compared as both continuous and categorical variables. Age was grouped as ≤64, 65–74, or ≥75 years. Tumor length was grouped as ≤5 or ≥5 cm. KPS score was grouped as 70–80 or 90–100. Number of lymph node resection was grouped as ≤15 or >15. The disease location was categorized as upper (from thoracic inlet to level of tracheal bifurcation; 18–23 cm from incisors), middle (from tracheal bifurcation midway to gastroesophageal junction; 24–32 cm from incisors), or lower (from midway between tracheal bifurcation and gastroesophageal junction to gastroesophageal junction, including abdominal esophagus; 32–40 cm from incisors). Surgical margin was categorized as negative (no cancer at resection margin) or positive (microscopic or macroscopic residual cancer or M1). The histologic differentiation was categorized as well, moderate, or poor. Clinicopathological information was obtained from medical records and pathology reports.

Posttreatment surveillance included routine clinical and laboratory examinations. The predominant imaging method was computed tomography (CT). For patients suspicious for relapse or progression in CT scan, positron emission tomography–computed tomography (PET-CT) was strongly suggested to make a definite diagnosis. Follow-up evaluations were performed every 3 months during the first 2 years, every 6 months in the next 3 years, and annually thereafter. Treatment response was evaluated based on the Response Evaluation Criteria in Solid Tumors (version 1.1) [20]. Dates of death were obtained from clinical records, telephone calls to their relatives, or the central registry of Chinese Bureau of Population Statistics.

### 2.3. Nomogram Construction

Prognostic nomogram was constructed using the primary cohort. Clinicopathological factors with a *p*-value < 0.05 in the univariable Cox regression analysis were screened. Furthermore, previously reported characteristics that were significantly associated with long-term survival were also considered. Screened variables were as follows: demographics (age and gender), KPS, pathologic characteristics (location, length, T stage, N stage, differentiation, surgical margin, lymphovascular invasion, nerve invasion, and number of lymph node resection), and adjuvant therapy. The selected variables were included in the least absolute shrinkage and selection operator (LASSO) regression algorithm. Cross-validation was used to confirm suitable tuning parameter lambda (λ) for LASSO logistic regression [21]. Then, the most significant variables selected by LASSO were used for multivariable Cox proportional hazard analysis to predict the 5-year OS rate.

### 2.4. Nomogram Assessment and Validation

#### 2.4.1. Discrimination

The discriminative ability of the nomogram to predict 5-year OS was assessed using area under the curve (AUC) of receiver operating characteristic curve (ROC) [22]. AUC was calculated with the Cox regression model method. The AUC was between 0.5 and 1.0, indicating a decent level of discriminative ability for the nomogram, while 0.5 indicated a random outcome. C-index, which is appropriate for censored data, was also used for evaluating the discrimination [23]. The nomogram was also compared with TNM stage in discrimination to assess whether nomogram could provide more accurate survival prediction than the 8th AJCC TNM stage. Kaplan–Meier survival curves were constructed according to nomogram-based risk stratifications.

#### 2.4.2. Calibration

The calibration of the nomogram was assessed by calibration plot. Nomogram-predicted and actual 5-year OS rates were plotted and compared to further verify the predictive performance of the nomogram. 

#### 2.4.3. Internal and External Validations

Analyses on discrimination and calibration were performed in internal validation cohort of Chengdu and external validation cohort of Beijing. The proposed nomogram from primary cohort was used to assess each patient in validation cohorts.

### 2.5. Statistical Analysis

In the retrospective cohort, the primary endpoint OS was calculated from the date of diagnosis to death from any cause or censored at the date of last follow-up for patients who were alive. For RCT, OS was defined as the interval from surgery to death or censorship, whichever occurred first. Survival curves were constructed using the Kaplan–Meier method and compared by log-rank test [24]. Results of the Cox regression analysis and Kaplan–Meier curve were summarized as hazard ratio (HR), 95% confidence interval (CI), and *p*-value. In the comparison of clinicopathological characteristics, Student’s *t*-test and chi-square (χ^2^) test were applied for continuous variables and categorical variables, respectively. R statistical software (R: A Language and Environment for Statistical Computing. R Foundation for Statistical Computing, version 3.3.2, Vienna, Austria. ISBN 3-900051-07-0, URL http://www.R-project.org/, accessed on 08 November 2022) was used to perform the statistical analyses. LASSO regression analysis was operated with the “glmnet” package. Data visualization was performed using the ggplot2 package. Statistical significance was set as *p* < 0.05 in a two-tailed test. 

## 3. Results

After excluding patients with T4b (*n* = 53), cervical EC (*n* = 37), adenocarcinoma (*n* = 36), small cell carcinoma (*n* = 25), neuroendocrine carcinoma (*n* = 21), distant metastasis (*n* = 8), neoadjuvant treatment (*n* = 12), and adjuvant radiotherapy (*n* = 40), a total of 1795 patients in Sichuan Cancer Hospital were included in the statistical analysis and randomly divided into primary cohort (*n* = 1077) and internal validation cohort (*n* = 718) (Appendix A). For independent external validation, patients receiving esophagectomy alone (*n* = 54) or esophagectomy plus adjuvant chemoradiotherapy (*n* = 64) in the RCT were enrolled. 

The baseline clinicopathological characteristics of 1077 patients in the primary cohort, 718 patients for internal validation, and 118 patients for external validation were compared in Table 1. For the entire cohort (1913 patients), there were more males than females (ratio, 4.9:1). The median age was 62 years (range, 34–85 years). Most patients presented with moderate performance status (KPS, 70–80) (56.6%), middle thoracic tumor (51.7%), tumor length ≤5 cm (64.2%), T3-4a (83.2%), N0-1 (68.6%), negative surgical margin (95.5%), number of lymph node resection >15 (73%), moderate to poor differentiation (82.8%), negative lymphovascular invasion (79.4%), and negative nerve invasion (77.9%). After esophagectomy, most patients received no adjuvant treatment (48.8%), followed by chemotherapy (34.6%) and chemoradiotherapy (16.6%). All clinicopathological characteristics except lymphovascular invasion were similar between primary cohort and internal validation cohort (*p* > 0.05). Most of characteristics were significantly different between primary cohort and external validation cohort (*p* < 0.05) (Table 1).

In the primary cohort, the median follow-up time was 5.2 years for surviving patients. The median OS and 5-year OS rate were 3.8 years and 45.9%, respectively (Appendix A).

We applied a LASSO regression algorithm for variable selection. The tuning parameter λ for LASSO regression was 0.003 when the partial likelihood binomial deviance was at its minimum (Figure 1A). The LASSO analysis retained eight variables with nonzero coefficients (Figure 1B). Independent prognostic factors concluded from LASSO analysis were as followings: gender, pathological T stage, pathological N stage, tumor differentiation, surgical margin, lymphovascular invasion, number of lymph node resection, and adjuvant therapy. A nomogram to predict 5-year OS was developed using clinicopathological factors above (Figure 2). The detailed nomogram score formula is presented in Appendix A.

In primary cohort, discriminative ability for 5-year OS measured by AUC and C-index was 0.76 (95% CI, 0.73–0.80) and 0.68 (95% CI, 0.66–0.71), respectively (Figure 3A). In the calibration plot for survival probability, nomogram-predicted and actual 5-year OS rates were highly correlated (Figure 3B). 

The AUC for discrimination in internal and external validations was 0.79 (95% CI, 0.75–0.83) and 0.77 (95% CI, 0.66–0.88), respectively. The C-index in internal and external validations was 0.67 (95% CI, 0.64–0.70) and 0.71 (95% CI, 0.63–0.79), respectively. These results indicated consistent and good discriminative ability (Figure 3C,E). Nomogram-predicted 5-year OS showed an optimal consistency with actual 5-year OS rate in validations (Figure 3D,F).

To further evaluate the discriminative ability, we plotted Kaplan–Meier curves according to risk stratifications by the quartiles of nomogram scores in primary cohort: low risk (nomogram score ≤140, *n* = 269), intermediate risk (nomogram score 141–167, *n* = 269), high-intermediate risk (nomogram score 168–195, *n* = 269), and high risk (nomogram score ≥ 196, *n* = 270). Compared with the low-risk group, patients with intermediate risk (HR, 2.45; 95% CI, 1.81–3.31; *p* < 0.001), high-intermediate risk (HR, 3.42; 95% CI, 2.55–4.59; *p* < 0.001), and high risk (HR, 6.46; 95% CI, 4.86–8.58; *p* < 0.001) had a substantially worse survival. The 5-year OS rates were 75%, 50.4%, 38.6%, and 18% for patients with low, intermediate, high-intermediate, and high risk, respectively (Figure 4A).

Validation cohorts were also stratified into four risk groups according to the same criteria. In the internal validation cohort, when compared with the low-risk group, a significantly worse survival was also observed in intermediate-risk (HR, 1.37; 95% CI, 1.01–1.93; *p* = 0.047), high-intermediate-risk (HR, 2.09; 95% CI, 1.5–2.91; *p* < 0.001), and high-risk (HR, 4.07; 95% CI, 3.01–5.49; *p* < 0.001) groups, with 5-year OS rate of 66.7%, 58.4%, 42.5%, and 14.4%, respectively (Figure 4C). Due to a limited number of patients in the external validation cohort, only the high-risk group showed a significantly worse prognosis than the low-risk group (HR, 3.24; 95% CI, 1.34–7.85; *p* < 0.01) (Figure 4E). The internal and external validations showed acceptable consistency with the primary cohort in nomogram-based risk stratifications. 

Compared with 8th AJCC TNM stage, the nomogram displayed better accuracy for predicting survival in the primary cohort (AUC: 0.76 vs. 0.72, *p* < 0.01; C-index: 0.68 vs. 0.65, *p* < 0.05), internal validation cohort (AUC: 0.79 vs. 0.71, *p* < 0.01; C-index: 0.67 vs. 0.62, *p* < 0.01), and external validation cohort (AUC: 0.77 vs. 0.69, *p* < 0.01; C-index: 0.71 vs. 0.62, *p* < 0.01). The TNM stage was unsatisfactory for the stratification of patients with stage IIA or IIB in the primary cohort (*p* = 0.06) (Figure 4B); stage IIA, IIB, or IIIA in the internal validation cohort (stage IIB vs. IIA, *p* = 0.53; stage IIIA vs. IIA, *p* = 0.18) (Figure 4D); and stage IIB or IIIA in the external validation cohort (*p* = 0.28) (Figure 4F). The nomogram showed a better discriminative accuracy than 8th AJCC TNM in OS prediction for patients with LA-ESCC.

According to the proposed nomogram, after radical surgery, oncologists and patients could determine the 5-year OS rate only by choosing different adjuvant therapies. After assuming that all patients had undergone surgery alone (total point started from 31.7), the predicted nomogram score and its corresponding 5-year OS (value α) were calculated. After the reception of adjuvant chemoradiotherapy or chemotherapy, a new nomogram score and its corresponding 5-year OS (value β) were obtained again. Improvements of 5-year OS was equal to the value β minus α. Then, we plotted a curve showing 5-year OS improvements attributing to adjuvant chemoradiotherapy or chemotherapy corresponding to nomogram scores (Y axis: β—α; X axis: nomogram total point). Patients with nomogram total points ranging from 110 to 260 (nearly all patients) were estimated to achieve at least 10% improvement in 5-year OS rate after receiving adjuvant chemoradiotherapy (Figure 5A). Meanwhile, after the reception of adjuvant chemotherapy, at least 10% improvement in 5-year OS rate was expected in patients with nomogram total points 159 to 207 (mainly high-intermediate-risk subgroup) (Figure 5B). Compared with chemotherapy alone, the addition of adjuvant radiotherapy further improved survival, indicating the important role of adjuvant radiotherapy in LA-ESCC after esophagectomy.

## 4. Discussion

This study enrolled a large scale of Chinese patients to develop a prognostic nomogram for LA-ESCC after esophagectomy and guide risk-dependent adjuvant therapies. With a median follow-up more than 5 years, a prognostic nomogram with satisfactory discrimination and calibration was developed based on eight significant clinicopathological factors, with comprehensive internal and external validations. Compared with the 8th AJCC TNM stage, a significant improvement of 4–8% in predictive accuracy was achieved by the nomogram. Survival benefits from adjuvant chemoradiotherapy/chemotherapy were quantified corresponding to nomogram scores, and risk-dependent and individualized adjuvant treatment options were recommended to patients. These findings provided new evidence supporting the use of nomogram for survival prediction and guiding precise adjuvant therapy in patients with LA-ESCC after esophagectomy.

Consistent with previous findings [10,15,16,17], the most significant prognostic factor in the nomogram was pathological T stage, followed by pathological N stage and adjuvant chemoradiotherapy, indicating the important role of adjuvant chemoradiotherapy in survival improvement. For example, when other clinicopathological factors were the same, nomogram scores of patients with pT3N0M0 plus adjuvant chemoradiotherapy were even lower than those with pT2N0M0 plus observation, demonstrating that locally advanced disease with effective postoperative chemoradiotherapy has greater potential to show a non-inferior survival than early disease. For patients with LA-ESCC, adjuvant radiotherapy could significantly increase local control rate and decrease local recurrences, which were ultimately converted to prolonged OS [4,25]. In the prospective RCT [4], compared with surgery alone, the addition of postoperative concurrent chemoradiotherapy and radiotherapy in LA-ESCCs significantly improved 18.5% and 12.8% in the 3-year OS rate, respectively. An improvement of 12.8% and 5.7% in the 3-year OS was attributed to radiotherapy and chemotherapy, respectively, which indicated adjuvant radiotherapy has a more important role than chemotherapy. This real-world retrospective study further confirmed that the addition of adjuvant radiotherapy to chemotherapy brought an increase of 10–20% in 5-year OS rate in most cases. In contrast, adjuvant chemotherapy alone could only bring up to 10% improvement in the 5-year OS rate in a small proportion of patients. Furthermore, only concurrent chemoradiotherapy and not chemotherapy alone could significantly improve long-term survival for patients with low to intermediate prognostic risk (nomogram score 110 to 159) (Figure 5A,B).

Currently, therapeutic guidelines manage and recommend adjuvant treatments mainly based on pathological TNM stage. Although other prognostic factors are also considered in decision making, the quantitative relationship between these prognostic factors and long-term survival is not always clear [26]. After the successful development of a nomogram, survival improvement attributing to each prognostic factor could be easily calculated and quantified. Most importantly, considering that adjuvant therapy was the only factor determined by oncologists, we provided an intuitive and visual survival improvement curve corresponding to each nomogram score after the reception of adjuvant therapy. Direct and clear clinical evidence are provided to oncologists for a reasonable decision on individualized treatment. Furthermore, the data visualization of risk-dependent survival improvement also makes health information more accessible to patients, reducing the communication barrier between oncologists and patients.

The strengths of this study included large sample size, comprehensive validation, and risk-dependent adjuvant therapy recommendation. First, nearly 2000 LA-ESCC patients with a median follow-up more than 5 years were enrolled. The prognostic nomogram based on this large-scale cohort was believed to provide precise and practical survival prediction for Chinese patients. Second, internal and external validations were comprehensively performed. Almost equivalent accuracy and reliability of the nomogram were concluded after external validation in an independent population with totally different characteristics, suggesting a good generalizability for different patients. Third, quantified survival benefits after adjuvant therapy were easily estimated, and risk-dependent and individualized adjuvant treatment recommendations could be offered to patients according to their nomogram scores. Fourth, this real-world data-derived nomogram had significant and practical value for decision making in the clinic. The easy availability of these prognostic factors from medical record also facilitated its application. 

This study should be considered in the context of certain weaknesses. First, this was a retrospective study with certain biases or confounders. Although multivariable analysis was performed in this study, biases coexisted with data and were difficult to remove radically by statistical optimization. The retrospective nature of this study should be considered when interpreting these results. Second, this study was based on the data of a single institute in China, and therefore, external validation process was important to improve reliability of the conclusions. The small sample size of the external cohort weakened its robustness in validation. Third, although pre-operative KPS was balanced among groups and an insignificant role of pre-operative KPS was found in prognosis, the post-operative KPS score was not considered separately in this study. At our institution, adjuvant treatment was generally determined by the opinion of multidisciplinary team. Patients’ post-operative general condition might also affect the prognosis, and the lack of post-operative KPS should be regarded as one of potential biases. Fourth, caution should be exercised in applying this nomogram in patients of Western countries, in which the major pathological type, treatment pattern, and image approach in follow-up are different from this study. The predominant imaging method is CT in China, while PET-CT is nowadays widely used in follow-up in Western counties [27]. Fifth, clinicopathological factors on the molecular or genetic level were absent in this study, and explorations on gene mutation profile, tumor microenvironment, and signaling pathways are suggested for further research [28,29,30,31].

## 5. Conclusions

In conclusion, we have developed and validated a nomogram that can predict 5-year OS for LA-ESCC after esophagectomy with a high degree of accuracy based on a large-scale cohort in China. The proposed nomogram shows better performance than the current TNM stage and provides risk-dependent and individualized adjuvant treatment options for Chinese patients.

## Figures and Tables

**Figure 1 cancers-14-05567-f001:**
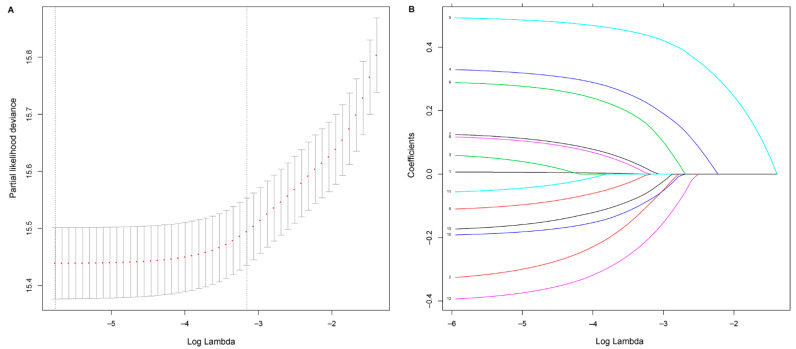
Selection of prognostic factors using least absolute shrinkage and selection operator (LASSO) Cox regression. (**A**) Tuning parameter λ based on minimum criteria in the LASSO regression. The partial likelihood binomial deviance is plotted against log λ. Using the minimum criteria and the one standard error of the minimum criteria, dotted vertical lines are set at the optimal values log λ, where factors are selected. (**B**) For clinicopathological features, LASSO coefficient profiles are plotted vs. log λ sequences. The dotted vertical line shows the nonzero coefficients, where eight nonzero coefficients are included.

**Figure 2 cancers-14-05567-f002:**
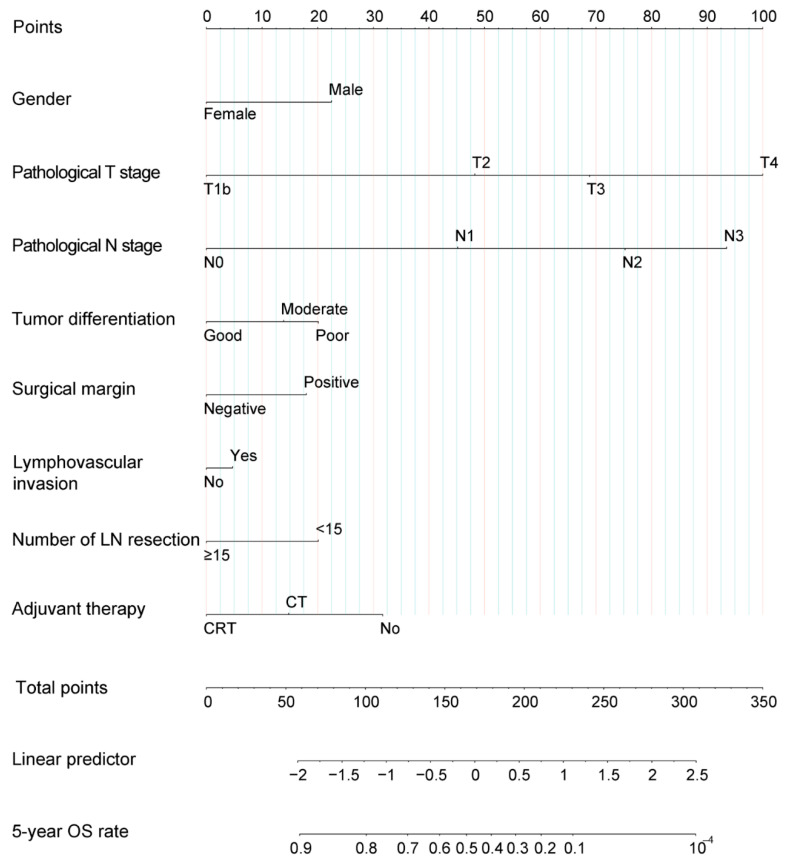
Nomogram for patients with locally advanced esophageal squamous cell carcinoma after esophagectomy. For each variable, an individual’s value is placed on the axis, and a line is drawn upward to determine how many points for each variable. The survival axis is drawn below the total points axis, which is then used to determine 5-year OS rates. LN, lymph node; OS, overall survival.

**Figure 3 cancers-14-05567-f003:**
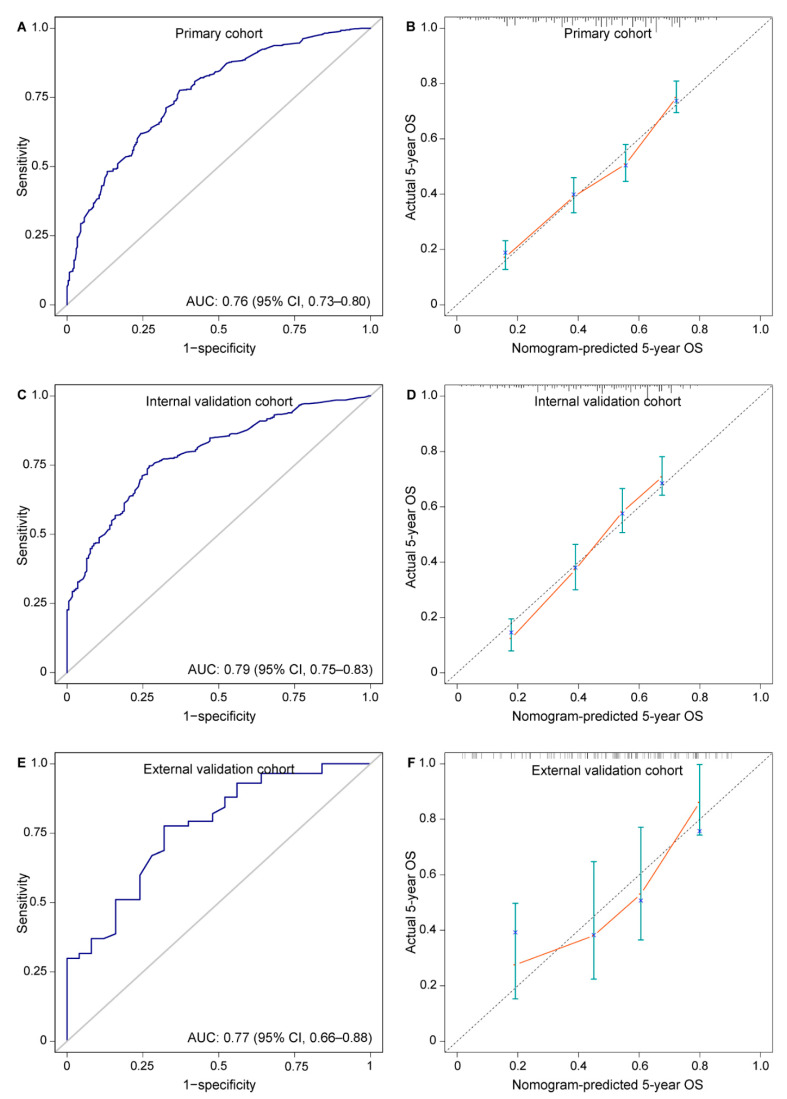
Assessment of nomogram predicting 5-year OS rate in patients with locally advanced esophageal squamous cell carcinoma after esophagectomy. (**A**) AUC 0.76 (95% CI, 0.73–0.80) in primary cohort; (**B**) calibration plot for the prediction of 5-year OS in primary cohort; (**C**) AUC 0.79 (95% CI, 0.75–0.83) in internal validation cohort; (**D**) calibration plot for the prediction of 5-year OS in internal validation cohort; (**E**) AUC 0.77 (95% CI, 0.66–0.88) in external validation cohort; (**F**) calibration plot for the prediction of 5-year OS in external validation cohort. Nomogram-predicted 5-year OS is plotted on the X axis; actual 5-year OS is plotted on the Y axis. AUC, area under the curve; OS, overall survival.

**Figure 4 cancers-14-05567-f004:**
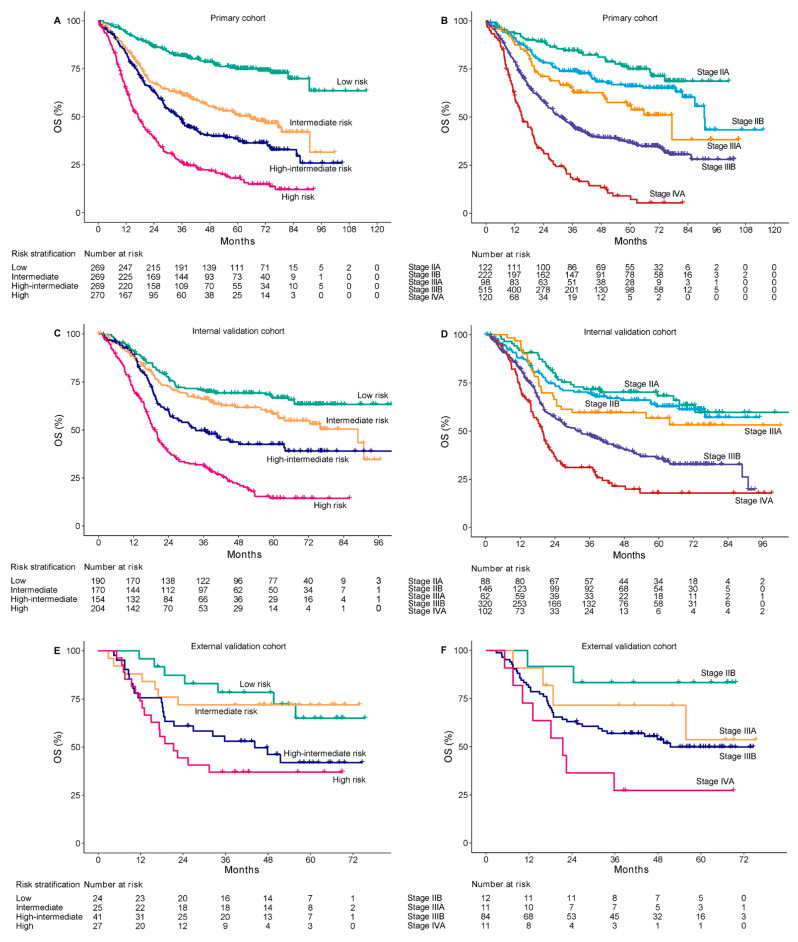
Kaplan–Meier overall survival curves of according to the quartile stratification of nomogram score and 8th AJCC TNM stage. The primary cohort (**A**,**B**), internal validation cohort (**C**,**D**), and external validation cohort (**E**,**F**). OS, overall survival.

**Figure 5 cancers-14-05567-f005:**
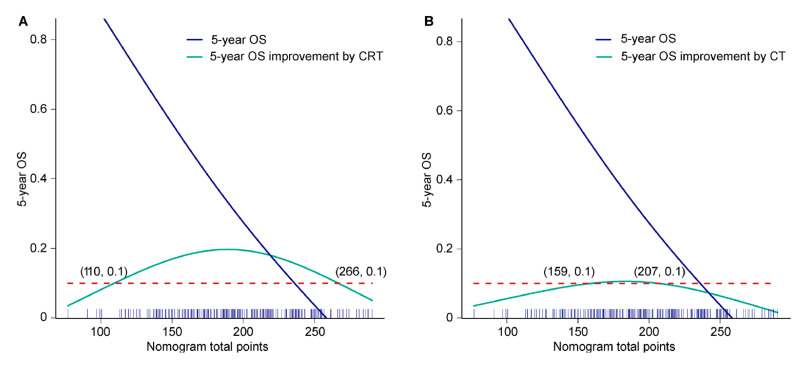
The 5-year OS improvement corresponding to the nomogram total point after (**A**) adjuvant chemoradiotherapy and (**B**) adjuvant chemotherapy. Blue solid line, 5-year OS rate corresponding to nomogram total point; green solid line, 5-year OS rate improvement (β—α) attributing to adjuvant therapy corresponding to nomogram total point; red dotted line, 5-year OS equal to 10%; blue stick on the X axis, the distribution frequency of nomogram total point. CRT, chemoradiotherapy; CT, chemotherapy; OS, overall survival.

**Table 1 cancers-14-05567-t001:** Clinicopathological characteristics.

Characteristic	All Patients No. (%)	Primary Cohort No. (%)	Internal Validation Cohort No. (%)	External Validation Cohort No. (%)	*p **	*p ^#^*
Total	1913	1077	718	118		
Age (years)						
As continuous variable					0.921	0.082
As categorical variable					0.892	0.002
≤64	1312 (68.6)	725 (67.3)	491 (68.4)	96 (81.4)		
65–74	506 (26.5)	294 (27.3)	190 (26.5)	22 (18.6)		
≥75	95 (4.9)	58 (5.4)	37 (5.2)	0 (0)		
Gender					0.151	0.025
Male	1589 (83.1)	878 (81.5)	605 (84.3)	106 (89.8)		
Female	324 (16.9)	199 (18.5)	113 (15.7)	12 (10.2)		
KPS						
As continuous variable					0.487	0.685
As categorical variable					0.411	0.733
70–80	1083 (56.6)	620 (57.6)	397 (55.3)	66 (55.9)		
90–100	830 (43.4)	457 (42.4)	321 (44.7)	52 (44.1)		
Tumor location					0.352	<0.001
Upper	460 (24.0)	281 (26.1)	173 (24.1)	6 (5.1)		
Middle	989 (51.7)	571 (53)	376 (52.4)	42 (35.6)		
Lower	464 (24.3)	225 (20.9)	169 (23.5)	70 (59.3)		
Tumor length (cm)						
As continuous variable					0.235	0.792
As categorical variable					0.145	0.874
≤5	1225 (64.2)	674 (62.8)	476 (66.3)	75 (63.6)		
>5	684 (35.8)	399 (37.2)	242 (33.7)	43 (36.4)		
Pathological T stage					0.436	<0.001
T1b	73 (3.8)	34 (3.2)	23 (3.2)	16 (13.6)		
T2	248 (13.0)	145 (13.5)	88 (12.3)	15 (12.7)		
T3	1415 (74.0)	806 (74.8)	530 (73.8)	79 (66.9)		
T4a	177 (9.2)	92 (8.5)	77 (10.7)	8 (6.8)		
Pathological N stage					0.352	<0.001
N0	595 (31.1)	349 (32.4)	242 (33.7)	4 (3.4)		
N1	717 (37.5)	401 (37.2)	247 (34.4)	69 (58.5)		
N2	407 (21.3)	226 (21)	146 (20.3)	35 (29.7)		
N3	194 (10.1)	101 (9.4)	83 (11.6)	10 (8.5)		
Surgical margin					0.975	0.016
Negative	1827 (95.5)	1026 (95.3)	683 (95.1)	118 (100.0)		
Positive	86 (4.5)	51 (4.7)	35 (4.9)	0 (0)		
Number of LN resection						
As continuous variable					0.835	<0.001
As categorical variable					0.724	<0.001
≤15	517 (27.0)	301 (27.9)	207 (28.8)	9 (7.6)		
>15	1396 (73.0)	776 (72.1)	511 (71.2)	109 (92.4)		
Tumor differentiation					0.667	<0.001
Well	329 (17.2)	187 (17.4)	135 (18.8)	7 (5.9)		
Moderate	819 (42.8)	458 (42.5)	293 (40.8)	68 (57.6)		
Poor	765 (40.0)	432 (40.1)	290 (40.4)	43 (36.4)		
Lymphovascular invasion					0.034	<0.001
Yes	395 (20.6)	188 (17.5)	155 (21.6)	52 (44.1)		
No	1518 (79.4)	889 (82.5)	563 (78.4)	66 (55.9)		
Nerve invasion					0.788	0.009
Yes	412 (22.1)	235 (21.8)	152 (21.2)	25 (35.2)		
No	1454 (77.9)	842 (78.2)	566 (78.8)	46 (64.8)		
Adjuvant therapy					0.184	<0.001
No	933 (48.8)	546 (50.7)	333 (46.4)	54 (45.8)		
Chemotherapy	662 (34.6)	381 (35.4)	281 (39.1)	0 (0)		
Chemoradiotherapy	318 (16.6)	150 (13.9)	104 (14.5)	64 (54.2)		

*p* * refers to *p*-value in the comparison between primary cohort and internal validation cohort. *p*
^#^ refers to *p*-value in the comparison between primary cohort and external validation cohort. Abbreviations: No., number; KPS, Karnofsky performance status; LN, lymph node.

## Data Availability

The data presented in this study are available on request from the corresponding author. The data are not publicly available due to privacy and ethical restrictions.

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
