# Peer review of "Nomogram-Based Survival Predictions and Treatment Recommendations for Locally Advanced Esophageal Squamous Cell Carcinoma Treated with Upfront Surgery"

_cancers, 2022, doi:10.3390/cancers14225567_

Round 1

Reviewer 1 Report

Thank you for the opportunity to review the paper entitled "Nomogram-Based Survival Predictions and Treatment Recommendations for Locally 2 Advanced Esophageal Squamous Cell Carcinoma". The analysis of a very large number of cases in this paper is highly commendable. Also, the perspective on risk classification by nomogram is interesting. However, several points should be revised.

Comments

1.       Regarding post-operative chemotherapy and post-operative chemoradiotherapy, how do you select the adjuvant treatment at your institution? Since esophageal cancer surgery is very invasive, not only the postoperative pathological findings but also the patient's postoperative general condition may affect the prognosis. I think it is necessary to mention this bias in the Discussion section. Please consider this.

2.       Figure 2; In the nomogram, the inequality symbols attached to the number of lymph node excisions are different from those in Table 1. Please check and amend it. Just to confirm, the more lymph nodes resected, the better the prognosis?

3.       Table 1; The first row of Table 1 is difficult to see, so please align it.

4.       Line 123; The histologic differentiation is commonly divided into well, moderate, and poor.

5.       Line 191; The font size of esophagectomy is large.

6.       Line 351; The word “date” may be “data”.

Author Response

Questions 1. Regarding post-operative chemotherapy and post-operative chemoradiotherapy, how do you select the adjuvant treatment at your institution? Since esophageal cancer surgery is very invasive, not only the postoperative pathological findings but also the patient's postoperative general condition may affect the prognosis. I think it is necessary to mention this bias in the Discussion section. Please consider this.

Answer: We deeply agree with the reviewer’s opinion. At our institution, adjuvant treatment was generally determined by the opinion of multidisciplinary team (MDT), mainly including surgeons, medical oncologist, and radiation oncologist. The pre-operative Karnofsky Performance Status (KPS) score was considered in this study (as listed in the Table 1). KPS scores were balanced among primary, internal validation, and external validation cohorts without significant differences in statistics. In the process of model development, we found that KPS was not an independent prognostic factor in the nomogram. In our institute, nutritional support was an important part in the treatment esophageal cancer. Malnutrition was required to be corrected as soon as possible, and patient education about nutritional support was also emphasized by nursing team. A poor KPS score might be improved in some degree after strong nutritional support, which may be one of potential explanations that KPS score was an insignificant prognostic factor in this study. In summary, considering the balanced relationship of pre-operative KPS among groups and insignificant role of pre-operative KPS in prognosis, the post-operative KPS score was not considered separately.

We agree with the reviewer’s opinion that postoperative general condition should be considered as one of bias. We added this as limitation in the Discussion. (Line 369-374, Page 14-15)

Questions 2. Figure 2; In the nomogram, the inequality symbols attached to the number of lymph node excisions are different from those in Table 1. Please check and amend it. Just to confirm, the more lymph nodes resected, the better the prognosis ?

Answer: Thanks for the reviewer’s careful check. There was an error in the Figure 2, and we have corrected it in the revised manuscript (Page 7).

Previously, many high-impact studies have confirmed that the number of lymph nodes removed is an independent predictor of survival after esophagectomy for esophageal cancer, and a higher nodal count favorably influences survival1,2. The same result was concluded in our study.

Question 3. Table 1; The first row of Table 1 is difficult to see, so please align it.

Answer: We aligned the first row of Table 1 according to the reviewer’s suggestion.

Question 4. Line 123; The histologic differentiation is commonly divided into well, moderate, and poor.

Answer: Thanks for reviewer’s suggestion. We make corresponding corrections in the manuscript (Line 129, Page 3), Table 1, and Figure 2 according to the reviewer’s suggestion.

Question 5. Line 191; The font size of esophagectomy is large.

Answer: We corrected the error in the revised manuscript (Line 199, Page 4) according to the reviewer’s suggestion.

Question 6. Line 351; The word “date” may be “data”.

Answer: Thanks. We corrected “date” to “data” in the revised manuscript (Line 366, Page 14).

References

  1. Altorki NK, Zhou XK, Stiles B, et al. Total number of resected lymph nodes predicts survival in esophageal cancer. Ann Surg. 2008;248(2):221-226.
  2. Peyre CG, Hagen JA, DeMeester SR, et al. The number of lymph nodes removed predicts survival in esophageal cancer: an international study on the impact of extent of surgical resection. Ann Surg. 2008;248(4):549-556.

Reviewer 2 Report

This study developed a nomogram for predicting the prognosis of locally advanced esophageal squamous cell carcinoma. The model was constructed on a large series of patients and was validated in an internal cohort as well as an external series. The nomogram showed good performance and indicated a significant survival improvement from adjuvant chemoradiotherapy in patients with high and intermediate risk.

The manuscript is very well written. The conclusions are consistent with the data presented.

In Figure 4, in panels B and D it would be better to exchange the colors of the curves stage IIIA and stage IIIB (blue and yellow), in order to be consistent with panels A, B, C and E.

Author Response

This study developed a nomogram for predicting the prognosis of locally advanced esophageal squamous cell carcinoma. The model was constructed on a large series of patients and was validated in an internal cohort as well as an external series. The nomogram showed good performance and indicated a significant survival improvement from adjuvant chemoradiotherapy in patients with high and intermediate risk.

The manuscript is very well written. The conclusions are consistent with the data presented.

Question 1: In Figure 4, in panels B and D it would be better to exchange the colors of the curves stage IIIA and stage IIIB (blue and yellow), in order to be consistent with panels A, B, C and E.

Answer: Thanks for reviewer’s suggestions. We exchanged the color of curves stage IIIA and IIIB of Figure 4B and D in the revised manuscript (Page 12).

Reviewer 3 Report

The paper entitled “Nomogram-based survival predictions and treatment recommendations for locally advanced esophageal squamous cell carcinoma” is a meaningful work investigating the efficacy of several clinical risk factors as prognostic tools in locally advanced esophageal squamous cell carcinoma (LA-ESCC).

The text is clear to read, and the methods used were appropriate and described with plenty of details. Overall, this is a well-designed study with rigorous methods. The study is on a timely subject, demonstrating the potential of novel prognostic tools in esophageal cancer.

Major revisions:

1.     The paper focuses on patients who underwent esophagectomy with curative intention. Why did the authors not choose disease-free survival (DFS) as the primary outcome measure, which could better represent the expected survival benefit derived from the procedure?

2.     The authors selected adjuvant therapy (including CTR, CT, or none) as an independent prognostic factor in the nomogram. As commonsense, adjuvant therapy decision-making involves the other selected factors, such as TNM stage, tumor differentiation surgical margin, and lymphovascular invasion. It seems inappropriate that adjuvant therapy was listed as an independent prognostic factor in the nomogram. I suggest performing subgroup analyses according to adjuvant therapy.

3.     According to guidelines, neoadjuvant radiotherapy, neoadjuvant chemotherapy, and neoadjuvant chemoradiotherapy are recommended in patients classified as cT3 to cT4a (any N) before the procedure. The authors ignored the impact of neoadjuvant therapy and did not mention it in clinicopathological characteristics. If this is not necessary, please explain and discuss the reasons.

4.     Usually, tumor location and size are vital measures affecting the procedures and are closely related to the patients’ prognostic outcome. Should it be added to the model, although they did not meet the criteria in the LASSO regression algorithm?

5.     The sample size of the external validation is limited, and the baseline clinicopathological characteristics of those patients were significantly different from those of other cohorts. Could the author provide a more convincing result of external validation?

6.     The authors validated the benefit for the patients corresponding to the nomogram by assuming that all patients had undergone surgery alone, which is unrealistic and unethical. I suggest detailing the decision-making process of adjuvant therapy based on the nomogram, which can have various implications for clinical practice.

Minor revisions:

1.     A study flow diagram of enrollment and follow-up should be made.

2.     Please detail the randomization method and provide the clinical trial registration number referring to the external validation for convenience and standardization.

3.     Please clarify the definition of tumor location, including the exact distance to the incisors or the division based on the adjacent structure.

4.     I suggest detailing the statistics of those continuous variables in Table 1.

5.     I suggest also providing the concordance index (C-index) of the nomogram.

Author Response

The paper entitled “Nomogram-based survival predictions and treatment recommendations for locally advanced esophageal squamous cell carcinoma” is a meaningful work investigating the efficacy of several clinical risk factors as prognostic tools in locally advanced esophageal squamous cell carcinoma (LA-ESCC). The text is clear to read, and the methods used were appropriate and described with plenty of details. Overall, this is a well-designed study with rigorous methods. The study is on a timely subject, demonstrating the potential of novel prognostic tools in esophageal cancer.

Major revisions:

Question 1. The paper focuses on patients who underwent esophagectomy with curative intention. Why did the authors not choose disease-free survival (DFS) as the primary outcome measure, which could better represent the expected survival benefit derived from the procedure?

Answer: Thanks for reviewer’s comment. As well recognized, overall survival (OS) was regarded as the gold standard to assess survival benefit of cancer patients. In this study, the primary purpose is to predict the survival of LA-ESCC patients with radical surgery. We tend to believe that patients and doctors care more about OS than disease-free survival (DFS). It is more meaningful in clinical practice to tell patients how long they will survive (because patients always asked doctors the same question).

DFS was commonly taken as primary endpoint in small-scale cohort with short follow-up time, for the purpose of gathering more targeted events and finishing research as soon as possible to save money (for example, clinical trials). Furthermore, it was well recognized that the prolongation of DFS would not always transfers to a longer OS time. Many clinical trials reported positive results on DFS, but with negative OS benefit. For some cancers with effective salvage treatment after progression (long post-progression survival), DFS could not predict OS.

This retrospective cohort is a large-scale cohort with long follow-up period (median follow-up > 5 years). There were enough death events occurring in the duration of follow-up. Therefore, we could conclude convincing results on gold standard OS, and it is less attractive for us to focus on DFS.

Question 2. The authors selected adjuvant therapy (including CTR, CT, or none) as an independent prognostic factor in the nomogram. As commonsense, adjuvant therapy decision-making involves the other selected factors, such as TNM stage, tumor differentiation surgical margin, and lymphovascular invasion. It seems inappropriate that adjuvant therapy was listed as an independent prognostic factor in the nomogram. I suggest performing subgroup analyses according to adjuvant therapy.

Answer: Thanks for reviewer’s suggestion. For a real-world retrospective cohort, long-term survival was determined by many factors. One of the most important factors was treatment strategy. Firstly, we observed death events in this study, and then this death event has always been retrospectively influenced by adjuvant treatment (CTR, CT, or none). That’s to say, all outcome data was related to adjuvant treatments. Therefore, we must list adjuvant therapy as one of factors, because survivals were truly associated with it.

For patients with no adjuvant therapy, the survival was fully determined by the basic patient and pathological characteristics without the intervention of treatment. We can also figure out how patient and pathological characteristics influence survival in the subgroup of these patients without adjuvant therapy in the established model in this study.

We classified adjuvant treatments to CRT, CT, or none with different nomogram score. In certain degree, the classification of adjuvant treatment is one kind of subgroup analysis. That’s to say, the nomogram already contained subgroup analysis. Nomogram was determined by multivariance Cox subgroup analysis.

Question 3. According to guidelines, neoadjuvant radiotherapy, neoadjuvant chemotherapy, and neoadjuvant chemoradiotherapy are recommended in patients classified as cT3 to cT4a (any N) before the procedure. The authors ignored the impact of neoadjuvant therapy and did not mention it in clinicopathological characteristics. If this is not necessary, please explain and discuss the reasons.

Answer: This study only included patients with direct esophagectomy first, with or without post-operative adjuvant therapy. Patients receiving any neoadjuvant treatment before surgery were excluded. To avoid misunderstanding, we added neoadjuvant therapy in the exclusive criteria in the revised manuscript (Line 87-88, Page 2).

In China, official guidelines recommend neoadjuvant chemoradiotherapy followed by radical esophagectomy in the treatment of LA-ESCC. However, the fact is that only a part of patients received neoadjuvant chemoradiotherapy first. Especially in less developed area, many hospitals have no multidisciplinary teams for cancer treatment and lack radiotherapy facilities. Radiotherapy facilities was in shortage, and radiotherapy cannot be performed in many hospitals. Treatment strategy is often determined by surgeons themselves, not with radiation oncologists or medical oncologists together. Based on their personal experience, some surgeons think that esophagectomy difficulty and post-operative complications will increase after neoadjuvant chemoradiotherapy. The conclusion in this study is more useful for those patients who receive surgery first in China and other Asian countries.

Question 4. Usually, tumor location and size are vital measures affecting the procedures and are closely related to the patients’ prognostic outcome. Should it be added to the model, although they did not meet the criteria in the LASSO regression algorithm?

Answer: Thanks for reviewer’s comment. Clinicopathological factors with a P value < 0.05 in the univariable Cox regression analysis were screened in the LASSO regression algorithm. Besides, previously reported characteristics which were significantly associated with long-term survival were also considered. Screening factors included demographics (age and gender), KPS, pathologic characteristics (location, length, T stage, N stage, differentiation, surgical margin, lymphovascular invasion, nerve invasion, and number of lymph node resection), and adjuvant therapy. Independent prognostic factors concluded from LASSO analysis were as followings: gender, pathological T stage, pathological N stage, tumor differentiation, surgical margin, lymphovascular invasion, number of lymph node resection, and adjuvant therapy.

LASSO regression algorithm is not a highly sensitive screening method in statistics. Therefore, in this study, we did not get the conclusion that tumor location and length were independent factors, which was concluded in some other study with different statistical methods. They did not meet the inclusive criteria in the LASSO regression algorithm, so they were not included in this nomogram model. The nomogram was totally established by LASSO regression algorithm, therefore I am afraid it is not appropriate to include tumor location and length as factors in the model.

Question 5. The sample size of the external validation is limited, and the baseline clinicopathological characteristics of those patients were significantly different from those of other cohorts. Could the author provide a more convincing result of external validation?

Answer: Thanks for reviewer’s suggestion. This study was based on the date of a single institute. Therefore, validation process was important to improve reliability of the conclusions. In this study, internal validation was performed in a cohort with similar clinicopathological characteristics. In external validation, clinicopathological factors are significantly different between primary and external validation cohorts. The AUC for discrimination in external validations was 0.77 (95% CI, 0.66-0.88), which is almost the same as the AUC value 0.76 (95% CI, 0.73-0.80) in the primary cohort. The results concluded from primary cohort were successfully validated in an independent cohort with a totally differently baseline factors. Therefore, we tend to believe the different characteristics between primary and external validation cohorts is one of major strengths, which suggested a good generalization of the predictive model.

Indeed, the small sample size in external validation cohort is one of the main limitations in this study. The RCT is a high-quality prospective cohort, with comprehensive quality control and follow-up. The results of this RCT have been published in the journal Oncologists (2021;26:e2151-60)1. We stated about the small sample size in external validation cohort as limitation in the Discussion (Line 368-369).

Question 6. The authors validated the benefit for the patients corresponding to the nomogram by assuming that all patients had undergone surgery alone, which is unrealistic and unethical. I suggest detailing the decision-making process of adjuvant therapy based on the nomogram, which can have various implications for clinical practice.

Answer: We appreciated the reviewer’s good suggestion. The conclusion of this study is only suitable for patients with direct surgery. One major clinical meaning of this study is mainly about the visualization of survival improvement by adding post-operative adjuvant therapy.

After radical surgery, oncologists and patients could determine the 5-year OS rate only by choosing different adjuvant therapies. After assuming that all patients had undergone surgery alone (total point started from 31.7), predicted nomgram score and its corresponding 5-year OS (value α) were calculated. After the reception of adjuvant chemoradiotherapy or chemotherapy, a new nomogram score and its corresponding 5-year OS (value β) were obtained again. Improvements of 5-year OS was equal to the value β minus α. Then, we plotted a curve showing 5-year OS improvements attributing to adjuvant chemoradiotherapy or chemotherapy corresponding to nomogram scores (Y axis: β – α; X axis: nomogram total point).

We added the detailed process of calculation about the survival improvement attributing to adjuvant therapy (Line 284-289, Page 13). We hope this would help readers to better understanding the concept.

Minor revisions:

Question 1. A study flow diagram of enrollment and follow-up should be made.

Answer: A flow chart was added as Supplemental Figure 1 according to the reviewer’s requirement.

Question 2. Please detail the randomization method and provide the clinical trial registration number referring to the external validation for convenience and standardization.

Answer: We added randomization method and clinical trial registration number as the reviewer’s requirement (Line 97-98, 101-102, Page 3).

Question 3. Please clarify the definition of tumor location, including the exact distance to the incisors or the division based on the adjacent structure.

Answer: We added the definition of tumor location in the revised manuscript according to the reviewer’s requirement (Line 122-126, Page 3).

Question 4. I suggest detailing the statistics of those continuous variables in Table 1.

Answer: We added statistical method for the comparison of continuous variables in the revised manuscript according to the reviewer’s requirement (Line 177-178, Page 4).

Question 5. I suggest also providing the concordance index (C-index) of the nomogram.

Answer: Thanks for reviewer’s suggestion. We added concordance index (C-index) in the revised manuscript. (Line 157-158, 231-237, 268-270).

References

  1. Ni W, Yu S, Xiao Z, et al. Postoperative Adjuvant Therapy Versus Surgery Alone for Stage IIB-III Esophageal Squamous Cell Carcinoma: A Phase III Randomized Controlled Trial. Oncologist. 2021;26(12):e2151-e2160.

Round 2

Reviewer 3 Report

Thank you for addressing all the questions.

I am satisfied with the authors’ reply to my comments, and I agree to publish this manuscript.

I hope the authors can add the content to the discussion for readers with the same concerns.

Author Response

We appreciate the reviewer's recommendation.